# Production of the light-activated elsinochrome phytotoxin in the soybean pathogen *Coniothyrium glycines* hints at virulence factor

Nicholas Greatens [1,2], Harun M. Murithi[3,4], Danny Coyne [4], Steven J. Clough[5,6], Michael Sulyok[7], Wahab Oluwanisola Okunowo[8], Hamed K. Abbas[9], W. Thomas Shier [10], Rachel A. Koch Bach [2]*

1 SCINet Program and ARS AI Center of Excellence, Office of National Programs, Agricultural Research Service, United States Department of Agriculture, Beltsville, Maryland, United States of America, 2 Foreign Disease-Weed Science Research Unit, Agricultural Research Service, United States Department of Agriculture, Fort Detrick, Maryland, United States of America, 3 Soybean Innovation Lab, University of Illinois, Urbana, Illinois, United States of America, 4 International Institute of Tropical Agriculture, Nairobi, Kenya, 5 Soybean/Maize Germplasm, Pathology, and Genetics Research Unit, Agricultural Research Service, United States Department of Agriculture, Urbana, Illinois, United States of America, 6 Department of Crop Sciences, University of Illinois, Urbana, Illinois, United States of America, 7 Department of Agricultural Sciences, Institute of Bioanalytics and Agro-Metabolomics, BOKU University - Universität für Bodenkultur Wien, Tulln, Austria, 8 Department of Biochemistry, College of Medicine, University of Lagos, Lagos, Nigeria, 9 Biological Control of Pests Research Unit, Agricultural Research Service, United States Department of Agriculture, Stoneville, Mississippi, United States of America, 10 Department of Medicinal Chemistry, College of Pharmacy, University of Minnesota, Minneapolis, Minnesota, United States of America

* rachel.kochbach@usda.gov

## Abstract

The Dothideomycete pathogen *Coniothyrium glycines* causes red leaf blotch of soybean, a major disease in Africa. It is one of two fungal plant pathogens on the USDA PPQ Select Agents and Toxins list of pathogens important to the biosecurity of the United States, reflective of its potential to be highly destructive if introduced. Despite its importance, there are no published reports regarding the molecular basis of host infection. Examination of the *C. glycines* genome revealed a secondary metabolite gene cluster that is similar in gene content and organization to clusters that synthesize light-activated perylenequinone toxins, such as cercosporin. Perylenequinones are non-host specific toxins that, upon exposure to light, generate reactive oxygen species, which have near-universal toxicity to plant hosts. *Coniothyrium glycines* isolates from eastern and southern Africa were cultured axenically under light and dark conditions. Light-grown cultures produced red-pink pigmentation typical of perylenequinones. Differential gene expression analysis showed that six of the eight genes in the biosynthetic gene cluster, including the polyketide synthase gene, were significantly upregulated in light. Liquid chromatography-mass spectrometry confirmed production of the perylenequinone elsinochrome A, a known virulence factor in other fungal pathogens. On leaves incubated in the dark, significantly fewer lesions formed and symptoms were delayed, compared to leaves incubated in the light. In addition,

**Data availability statement:** Genomic and transcriptomic data were deposited in the Sequence Read Archive (SRA) of the National Center for Biotechnology Information (NCBI) as part of projects PRJNA1226367 and PRJNA1226299, respectively. The annotated genome for LW.SEE.SI was deposited at NCBI GenBank under the accession JBMQJI000000000. All other data are available as supplementary files or at github.com/ngreatens/RLB_perylenequinone.

**Funding:** Financial support for this project was provided by USDA-ARS projects 8044-22000-051 (R.K.B.), 5012-22000-023 (S.C.), 6066-42000-007 (H.A.), 0201-88888-003 (R.K.B. and N.G.), and 0201-88888-002 (R.K.B. and N.G.), as well as the USDA-ARS National Plant Disease Recovery System (H.A. and W.T.S.).

**Competing interests:** The authors have declared that no competing interests exist.

we identified orthologous gene clusters in more distantly related Dothideomycete plant pathogens where their presence was previously unknown, indicating a broader importance of these toxins to agriculture and fungal ecology. This work provides the first evidence that elsinochrome A may contribute to the virulence of *C. glycines*.

## Introduction

*Coniothyrium glycines* (R.B. Stewart) Verkley & Gruyter, a Dothideomycete fungus within the Pleosporales, is the causal agent of red leaf blotch, a serious disease of soybean in Africa. Early in the infection process, characteristic red lesions are formed. As infection progresses, these lesions expand, leading to defoliation. Yield losses from *C. glycines* have been documented as high as 50% [1], and there are no reported sources of host resistance [2–4]. *Coniothyrium glycines* is one of two fungal species listed on the United States Department of Agriculture (USDA) Plant Protection and Quarantine Select Agents and Toxins list [5]. Select Agents, as defined by the USDA, are microorganisms that may pose a severe threat to public, animal or plant health or to animal or plant products. If *C. glycines* were to spread beyond Africa to major soybean producing regions of the world, it could cause devastating impacts to production [5,6]. While *C. glycines* is a plant pathogen of significant regulatory concern, there are no reports describing any of the molecular machinery associated with pathogenicity. Understanding this will be important for the development of management strategies that effectively control red leaf blotch.

An extensive array of secondary metabolites are produced by Dothideomycetes [7]. These secondary metabolites are small, often specialized, biologically active compounds that typically do not play a role in general metabolism, but which are important in fungal ecology, enabling organisms to outcompete others or to expand to new environmental niches [8]. Secondary metabolites have diverse uses in industry, medicine, agriculture, and food production and are frequent targets of genome mining. Identification of the genes responsible for production of these specialized metabolites is aided by their close physical linkage and organization into clusters [9]. With improvements in genome sequencing and advances in software for the prediction of biosynthetic gene clusters [10,11], the diversity, abundance and distribution of these specialized metabolic pathways is being brought into sharper focus.

Within the Dothideomycetes, many necrotrophic plant pathogenic species produce phytotoxic metabolites, or toxins, that are required for pathogenicity [7]. These toxins are typically classified based on their biological activity [7]. Host-specific toxins (HSTs) help determine the host range of the pathogen as they are biologically active against particular plant species [12]. Among the best studied HSTs are T-toxin, HC-toxin, and victorin, toxins produced by species of *Cochliobolus* that affect only some grass species [13]. Other toxins are non-host specific (non-HSTs) and are biologically active against a broad spectrum of plant species [14]. The best studied class of non-HSTs is the perylenequinones, which are produced by species within the Dothideomycetes [15], Sordariomycetes, and other Ascomycota [16].

Perylenequinone toxins are photoactivated, using light energy to generate singlet oxygen and free radical oxygen species. Activated oxygen species have near-universal toxicity because they damage essential molecules common to all cells [17]. Daub et al. [18] proposed that perylenequinone-producing fungi growing on leaves use these toxins to permeabilize adjacent cells, which then leak cellular contents that the fungi use for nutrition. Because light is required for the production and activation of these toxins, the severity of diseases caused by perylenequinone-producing fungi increases with light intensity and day length [19,20].

The best studied perylenequinone is cercosporin, first reported in the large Dothideomycete genus *Cercospora* [15]. Cercosporin, produced under optimal culture conditions and exposure to light, is apparent as a deep red pigment [21]. In plant lesions that result from cercosporin toxicity, reddish-purplish pigmentation is visible as a ring in the lesion margin [22]. Cercosporin is also produced by species of *Pseudocercosporella* [23] and *Colletotrichum* [24]. Genomic analyses show that biosynthetic clusters orthologous to those that produce cercosporin have been transferred horizontally multiple times and are now present in other important fungal plant pathogens such as *Parastagonospora nodorum*, cause of Septoria nodorum blotch of wheat, and *Fulvia fulvum* (syn. *Cladosporium fulvum)* [24]. Other fungi produce related perylenequinone toxins, all of which are thought to be preferentially synthesized upon exposure to light [17]. These toxins include altertoxin produced by species of *Alternaria* [25], elsinochromes by *Elsinoë* species and *Parastagonospora nodorum* [26,27] and hypocrellins by *Hypocrellin bambusae, Rubroshiraia bambusae* and *Shiraia* species [28–30].

Notably, the red lesions produced by *C. glycines* appear similar to lesions caused by perylenequinone toxicity, indicating a possible mechanism that enables this pathogen to cause disease. While several genomes are available for *C. glycines* [31], there has been no exploration of the biosynthetic gene clusters within the species. We assembled a draft genome of a contemporary *C. glycines* isolate collected from Zambia and identified its biosynthetic gene clusters. One cluster showed close homology to known perylenequinone-producing secondary metabolite gene clusters present in *Parastagonospora nodorum, Shiraia bambusicola,* and *Cercospora beticola,* which produce elsinochrome C, hypocrellin and shiraiachromes, and cercosporin, respectively. Since other perylenequinone-producing species produce a red-colored pigment when exposed to light [21], we grew eight *C. glycines* isolates under light and dark conditions and observed the same effect. Transcript profiling of cultures grown in the light confirmed significant upregulation of most genes in the gene cluster, and liquid chromatography-mass spectrometry identified elsinochrome A as a product, along with hypocrellin A to a lesser extent. Finally, we used detached leaf assays to compare infection under different lighting regimes. Notably, comparative genomics analyses revealed high homology of this gene cluster with those from several other fungi for which perylenequinone gene clusters have not been well-characterized or described, including lichenizing and endophytic fungi, as well as the important plant pathogen *Corynespora cassiicola,* a pathogen of rubber, cotton, and soybean, among other hosts. This is the first study to investigate secondary metabolites in *C. glycines* and their potential role in disease progression.

## Materials and methods

### Culturing

Eight geographically diverse *C. glycines* isolates were plated on potato dextrose agar (PDA) amended with chloramphenicol. Six plates per isolate were prepared. For each plate, a single sclerotium from stock isolates growing on 20% V8 agar was placed in the center. Three plates were maintained under a 12-hour light/dark cycle, while the other three were wrapped in aluminum foil to simulate 24-hour darkness. All plates were incubated at 24 °C-day and 20 °C-night. Cultures were imaged at 18 days. Collection information for the isolates used in this study is available in [32]. The initial infected leaf specimens were obtained through collaborations with agricultural institutions throughout Africa. Institutions granted permission for sampling, isolations, and subsequent research. More information is available upon request. The Foreign Disease-Weed Science Research Unit is a registered facility with the USDA APHIS Division of Agricultural Select Agents and Toxins Federal Select Agent Program (https://www.selectagents.gov), which allows the legal importation and maintenance of plant pathogen Select Agents.

## Genome sequencing

*C. glycines* isolate LW.SEE.SI, collected from diseased soybean near Lusaka, Zambia in 2023, was selected for genome sequencing. DNA was extracted following a previously described protocol [32], including an RNase treatment. DNA from LW.SEE.SI was cleaned with the Genomic DNA Clean & Concentrator kit (Zymo Research, Irvine, CA, USA). Library preparation and Illumina sequencing of 150 bp paired-end reads was conducted by Novogene Corporation Inc. with the Illumina NovaSeq X Plus sequencer at approximately $50_\times$ coverage. Genomic reads were deposited in the Sequence Read Archive (SRA) of the National Center for Biotechnology Information (NCBI) as part of project PRJNA1226367.

Raw reads were trimmed and filtered for quality using fastp [33] with the settings cut_front, cut_tail, detect_adapter_for_pe, and dedup enabled and default parameters. The quality of trimmed reads was assessed with FastQC [34] and MultiQC [35], summarizing fastp and fastqc reports. Reads were assembled with SPAdes v3.15.5 [36] using default settings, and scaffolds less than 1000 bp were discarded. The resulting assembly was assessed for contamination with BlobToolKit v4.3.5 [37] using Diamond [38] to query the UniProt database [39], BLASTn search [40] to query the NCBI nucleotide database, and BWA-MEM v2.2.1 [41] with default settings to map reads to scaffolds. After inspection of BLAST results and the gc-coverage plot, scaffolds with top hits to phyla other than Ascomycota were discarded. The assembly quality was assessed with BUSCO v5.7.0 [42] using the dothideomycetes_odb10 dataset downloaded 22-04-2024.

Repeats elements were modeled using RepeatModeler v2.0.4 with LTRStruct enabled [43] and masked with RepeatMasker v4.1.5 [44]. With the soft-masked assembly, BRAKER3 v3.0.3 [45] was used for gene prediction. Input included RNA sequencing (RNA-Seq) data for the LW.SEE.SI isolate under light and dark conditions (see below) with a dataset of genes present in >80% of 77 Dothideomycete genomes, as curated by OrthoDB v11 [46] and downloaded on 2024-05-13. The output gff3 file and assembly were submitted to the web version of AntiSMASH v7.0 [11] with all features enabled except cassis for biosynthetic gene cluster prediction. Output files from BRAKER3 were submitted to EggNOG 5.0 and interproscan v5.67-99 for functional annotation [47–49]. FunAnnotate v1.8.16 was used to prepare a file for submission [50]. The annotated genome has been deposited at NCBI GenBank under the accession JBMQJI000000000. The version described in this paper is version JBMQJI010000000.

## Phylogenetics and orthology analyses

A blastp search was conducted with the PKS gene of the putative perylenequinone gene cluster against the NCBI nr database. For each species with a match >60%, reference genomes were downloaded from NCBI with one species represented per genus (S1 Table). In addition, genomes for *Cercospora beticola, Colletotrichum fioriniae,* and *Fulvia fulvum*, were included since each is known to produce a perylenequinone toxin [24]. For genomes without annotations, genes were predicted using BRAKER3 as described above using the Dothidomycetes OrthoDB dataset as input. Orthology of PKS genes was inferred using Orthofinder [51]. Protein sequences of PKS genes orthologous with CTB1 (XP_023460065.1), in *C. beticola*, part of the cercosporin gene cluster, and its paralog XP_023450000.1, were aligned with MAFFT [52]. A maximum likelihood tree was inferred from the alignment with IQ-TREE 2 using the Q.yeast model and 1000 UltraFast bootstraps with the CTB1 paralog designated as an outgroup [53,54].

With the output from AntiSMASH, homologous gene clusters were further explored using the online tool CAGECAT [55], which employs the programs cblaster [56] and clinker [57] to visualize gene clusters beyond those included in the AntiSMASH databases. The tree and gene clusters were visualized together using the R packages ggtree [58] and gggenes [59]. Alignments and R code are available at github.com/ngreatens/RLB_perylenequinone.

## RNA sequencing and analysis

RNA sequencing (RNA-Seq) was conducted with eight *C. glycines* isolates (Table 1). For each isolate, approximately eight sclerotia from a dark-grown culture on 20% V8 media were plated on PDA amended with chloramphenicol layered

with a cellophane membrane (Gel Company, San Francisco, CA, USA) in 60 mm Petri dishes. Ten plates per isolate were prepared, with half grown under light conditions and half under dark in the same conditions as described above. After two weeks, tissue was scraped from the cellophane membranes and ground in liquid nitrogen. RNA was extracted with the E.Z.N.A. Fungal RNA Mini Kit (Omega BioTek, Norcross, GA, USA) with a compatible on-membrane DNase treatment (Omega BioTek, Norcross, GA, USA) and sequenced by Maryland Genomics (part of the University of Maryland School of Medicine Institute for Genome Sciences) on an Illumina NovaSeq 6000 sequencer with an S4 flow cell. Raw reads were deposited in the NCBI SRA as project PRJNA1226299 (S2 Table).

RNA-Seq reads were trimmed and filtered for quality using fastp with default settings, and quality assessed as described above. Trimmed reads were aligned to the curated LW.SEE.SI assembly with HiSat2 [60] with default settings, and mapped reads were sorted with SAMtools v1.16.1 [61]. Reads mapped to genes were quantified with feature-Counts, part of the R subread package (v2.0.4) [62], and counts were normalized with DESeq2 [63]. Genes were considered differentially upregulated if they had a log2-fold change greater than or equal to two and a false-discovery rate (FDR)-adjusted $p$ value < 0.05. All analyses for the packages listed above were conducted using R version 4.4.1 (http://www.r-project.org/).

## Liquid chromatography-tandem mass spectrometry

Liquid chromatography-tandem mass spectrometry (LC-MS/MS) was used to identify the pigmented compounds. Two plates for each of the eight isolates (Table 1) were prepared and grown under light and dark conditions as described above. After two weeks, tissue was scraped from a cellophane membrane, weighed, and placed in 1.5 mL cryovials with eight steel chrome beads and 1 mL of 50% methanol. Plugs of agar from below the growing tissue were removed with a 0.5 cm corer and were weighed and processed similarly. Finally, for three isolates—AN.T.W, EMETT.64 and KAB.10—we also sampled infected leaf tissue. Leaves infected in the lab (as described below) at 28 days post inoculation, as well as the original infected leaf specimens from which the original cultures were isolated, were analyzed [32]. Samples in cryovials were beat for two minutes with a FastPrep-24 Classic Bead Beating Grinder and Lysis System (MP Biomedicals, Irvine, CA, USA), centrifuged at max speed for 1 min, and the pigmented supernatant decanted. An additional 1 mL of 50% methanol was added to all samples in cryovials, and the beating/centrifuging/decanting process was repeated until no visible pigment was apparent in the supernatant, for a total of five times. The supernatant was filter-sterilized and dried in a desiccator. Dried samples were dissolved in 5 mL of acetonitrile/water/acetic acid (79/20/1, v/v/v) for 90 min on a rotary shaker, and an aliquot of 500 µL was diluted 1:1 with acetonitrile/water/acetic acid (20/79/1, v/v/v), of which 5 µL were injected into the LC-MS/MS system previously described [64]. A 1290 Series high performance liquid chromatography System (Agilent, Santa Clara, CA, USA) was coupled to a QTrap 5500 LC-MS/MS System (Applied Biosystems SCIEX, Framingham, MA, USA) outfitted with a Turbo Ion Spray electrospray ionization source. At 25°C, chromatographic separation was performed, running an acidified methanol/water gradient on a Gemini $C_{18}$-column, 150 × 46 mm i.d., 5 µm particle

**Table 1. Isolates used for RNA sequencing.**

| Isolate | Biosample | Country of origin | Year |
|---------|-----------|-------------------|------|
| AN.T.W | SAMN46927251 | Mozambique | 2023 |
| CHI.P67 | SAMN46927252 | Zimbabwe | 2023 |
| EMETT.64 | SAMN46927253 | Ethiopia | 2022 |
| EMETT.97 | SAMN46927254 | Ethiopia | 2022 |
| KAB.10 | SAMN46927255 | Uganda | 2021 |
| LW.SEE.SI | SAMN46927256 | Zambia | 2023 |
| MT.M.P17 | SAMN46927257 | Zambia | 2023 |
| NAM.2 | SAMN46927258 | Uganda | 2021 |

size, equipped with $C_{18}$ 4 × 3 mm i.d. security guard cartridge (Phenomenex, Torrance, CA, USA). The scheduled multiple reaction monitoring mode was used to acquire ESI-MS/MS data, both in positive and negative polarity in two separate chromatographic runs. Output was compared with a panel of mycotoxins and secondary metabolites, including cercosporin, as described in Sulyok et al. 2024 [64]. Standards of elsinochrome A (BioAustralis, Smithfield, Australia) and hypocrellin A and B (Ambeed, Arlington Heights, IL, USA) were also included in this panel. Final concentrations of metabolites were normalized to the mass of the tissue from which metabolites were extracted. For the two compounds detected in our samples—elsinochrome A and hypocrellin A—two *F*-tests of equality of variances were performed for each sample type (tissue and agar). Depending on the results, a *t*-test assuming equal or unequal variance was conducted for each sample type to determine whether the concentration of the compound produced per gram of tissue differed significantly between the light and dark treatments. A *P*-value < 0.05 was considered significant.

## Detached leaf assays

Detached leaf assays were used to establish whether RLB disease symptoms manifest similarly under light and dark conditions. Inoculum of isolates EMETT.97, MPO.P9, MT.M.P17 and NAL.3 was prepared as follows: five to eight sclerotia of the select isolate were placed on a 20% V8 plate covered with a cellophane membrane and grown for two weeks in the dark at 20°C. Mycelia and sclerotia were scraped from the cellophane, massed and 0.1 g was transferred to a 1.5 mL screw-top tube with six 1.6 mm sterilized chrome beads and 1 mL of sterilized water. Tubes were shaken at 6.5 m/s for five minutes using a FastPrep-24 Classic Bead Beating Grinder and Lysis System. 50 µL of slurry was added to 25 mL of water and the solution was transferred to a spray bottle.

Soybean cultivar Williams 82 was grown for 28 days under typical greenhouse conditions. Trifoliate and unifoliate leaves were harvested immediately before inoculation and kept within a folded moist paper towel while transporting to the lab (approximately 5 minutes). The base of each leaflet was cut with a razor to remove the petioles and improve water uptake, and ten leaflets were arranged on top of two layers of water-saturated paper towels in sterilized, clear plastic clamshell containers (13.6525 × 22.225 × 3.8 cm, Andex, Escanaba, MI, USA). Approximately 1 cm of each towel was folded over the cut leaflet bases to ensure contact between cut edges and the wet towels to enhance water uptake. Approximately 1.5 mL of the inoculum was sprayed evenly over the leaf surfaces and the boxes were closed and moved to incubation chambers. Two clamshells per isolate were incubated in a growth chamber: one was exposed to a 12-hour light/dark (24 °C/20 °C) cycle, while the other was wrapped in aluminum foil to simulate 24-hour darkness.

Leaves were monitored daily, and the following data were recorded: 1) days post inoculation (DPI) to symptom emergence for each leaf; and 2) number of lesions per leaf at 21 DPI. For each isolate and variable, *F*-tests of equality of variances were performed. Depending on the results, for each isolate, *t*-tests assuming equal or unequal variance were conducted to determine whether there were significant differences between lighting treatments in days to symptom emergence and the number of lesions. A *P*-value < 0.05 was considered significant.

## Results

### Pigment production in culture

In all isolates grown in light, there was visible pigment production after three days of growth. Most pigmentation production ceased after 18 days of growth (Fig 1). No pink-red pigmentation was observed in cultures that were maintained in 24-hour darkness (Fig 1).

### *Biosynthetic gene cluster content in LW.SEE.SI draft genome*

We sequenced a contemporary *C. glycines* isolate from Zambia, LW.SEE.SI. The final assembly was 32.2 Mb with 1181 scaffolds. 99.2% of Dothideomycete BUSCO genes were complete and present. AntiSMASH identified 41 biosynthetic gene clusters that were predicted to encode: eight nonribosomal peptide synthetases (NRPSs); three terpene synthases;

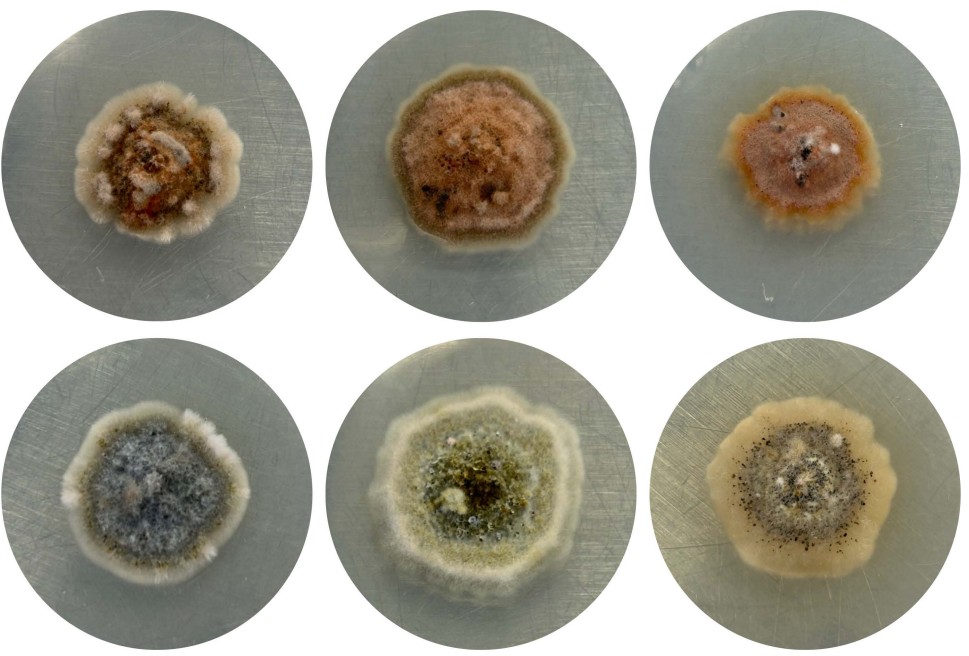

**Fig 1. *Coniothyrium glycines* isolates grown under light (top) and dark (bottom) conditions on PDA after 18 days.** From left to right: EMETT.97, KAB.10, LW.SEE.SI. Scale bar = 1 cm. Pigmentation is observed in cultures grown under light.

13 fungal RiPP ribosomally synthesised and post-translationally modified peptides (RiPPs); one cluster for indole biosynthesis; 12 polyketide synthases (PKS); and four hybrid NRPS/PKS clusters. Among the PKS clusters was a putative perylenequinone biosynthetic gene cluster, cluster 46.1, for which the KnownClusterBlast feature showed hits for elsinochrome and cercosporin biosynthetic clusters.

### Phylogenetic and orthology analysis of perylenequinone gene cluster

The PKS in cluster 46.1 had numerous blastp hits (>60% match) in the blast non-redundant proteins database to organisms in which perylenequinone toxin production has not previously been reported (e.g., *Paraphoma chrysanthemicola* and *Corynespora cassiicola*, another important pathogen of soybean, causing target spot), as well as others where it is well described (e.g., *Shiraia* spp., *Parastagonospora nodorum,* and *Elsinoe* spp.). A maximum likelihood tree shows that the *C. glycines* PKS gene is most closely related to homologs in *Shiraia* sp. slf14, *P. nodorum*, *P. chrysanthemicola,* and a species within the Phaeosphaeriaceae (Fig 2A). Exploration and visualization of the gene clusters using the online tool CAGECAT and the R packages ggtree and gggenes confirmed the presence of additional orthologous genes and conserved gene arrangement in these perylenequinone clusters of the species listed above (Fig 2A and Table 2). With these species, along with *C. cassiicola* and *Melanomma pulvis-pyrius*, eight core genes are conserved in arrangement and order in the perylenequinone biosynthetic gene cluster. In some cases, other genes are present within the cluster, including genes which are known to be involved in the biosynthesis of various other perylenequinones (e.g. *elcH*, colored brown in Fig 2). Notably, in *C. glycines,* as well as *P. chrysanthemicola* and the Phaeosphaeriaceae sp., an ortholog to *elcH* is present in the genome, but is located outside the biosynthetic cluster on another contig. The eight core genes are also present in two lichenizing fungi within Trypetheliaceae, *Bathelium mastoiderum* and *Viridothelium virens*, as well as *Elsinoë ampelina*, although gene order is different from *C. glycines* and *P. nodorum*. These PKSs form a clade that is sister to a clade composed of PKSs from multiple species, including *C. beticola*, *Colletotrichum fioriniae* and *Fulvia fulvum* (Fig 2A).

A

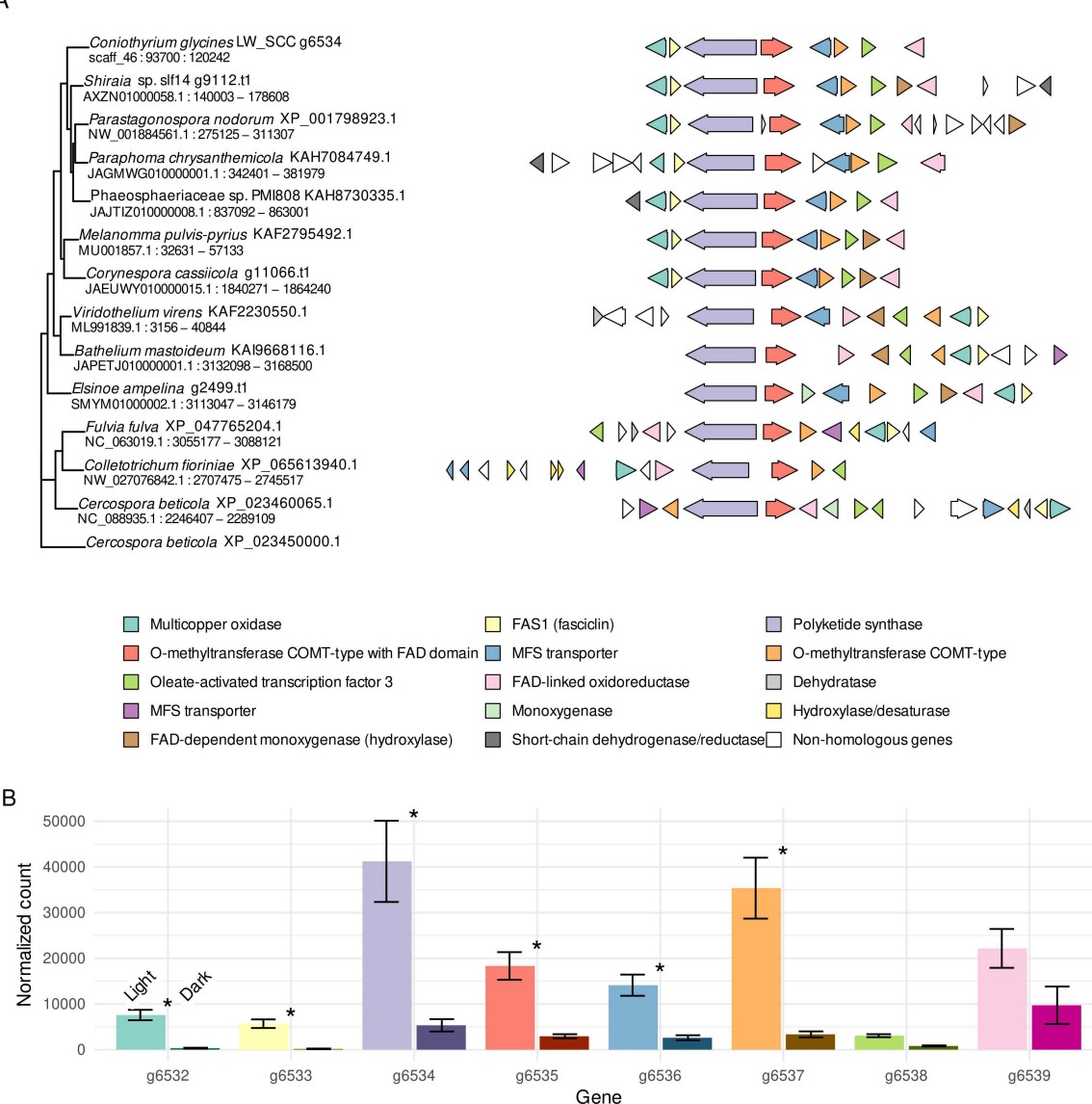

**Fig 2. Organization of perylenequinone biosynthetic gene cluster in *C. glycines* and other fungi and expression of cluster genes in *C. glycines* under light and dark conditions. A.** Predicted perylenequinone gene clusters in *Coniothyrium glycines* and twelve other species. Genes are colored based on homology to, in the order given in the legend, the eight genes within the *C. glycines* cluster, to *CTB4*, *CTB7*, *CTB9*, and *CTB10* (from *Cercospora beticola*), *elcH* (from *P. nodorum*), and *HYP12* (from *Shiraia* sp.), with functions based on InterPro [65] or annotations from NCBI as available (S3 Table). Clusters are arranged on branches of a maximum likelihood tree generated using the PKS genes (lavender). Cluster coordinates are given below the species and PKS protein names. Clusters vary in organization and presence of additional genes. **B.** Normalized counts of *C. glycines* transcripts of *g6532-g6539* averaged across eight samples in light and dark treatments. Error bars show one standard deviation, and significance (log2-fold change > 2; *P* < .05) is denoted with an asterisk. Genes are colored according to their function/identity, as shown in A, with level of expression in dark shown in a darker shade. Six of eight genes within the cluster are significantly upregulated under light conditions in *C. glycines*. Code for figures is available at github.com/ngreatens/RLB_perylenequinone.

## Gene expression

RNA-Seq analysis identified a total of 189 genes differentially expressed under differing light regimes, with 110 significantly upregulated in the dark treatment and 79 significantly upregulated in the light treatment (S4 Table). Six of the eight

**Table 2. Orthologs to genes in perylenequinone biosynthetic clusters from *Cercospora beticola* and *Parastagonospora nodorum*. (Updated NCBI gene IDs are given in parentheses).**

| Annotation | *Coniothyrium glycines* (elsinochrome) | *Cercospora beticola* (cercosporin) | *Parastagonospora nodorum* (elsinochrome) |
|---|---|---|---|
| Multicopper oxidase | *g6532 (ACN47E_006483)* | *CTB12* | *elcG* |
| Fasciclin | *g6533 (ACN47E_006484)* | *CTB11* | *elcF* |
| Polyketide synthase | *g6534 (ACN47E_006485)* | *CTB1* | *elcA* |
| O-methyltransferase COMT-type with FAD domain | *g6535 (ACN47E_006486)* | *CTB3* | *elcB* |
| MFS transporter | *g6536 (ACN47E_006487)* | --- | *elcC* |
| O-methyltransferase COMT-type | *g6537 (ACN47E_006488)* | *CTB2* | *elcD* |
| Oleate-activated transcription factor 3 | *g6538 (ACN47E_006489)* | *CTB8* | *elcR* |
| FAD-linked oxidoreductase | *g6539 (ACN47E_006490)* | *CTB5* | *elcE* |
| Hydroxylase | *g8166 (ACN47E_008109)\** | --- | *elcH* |
| MFS transporter | *g9122 (ACN47E_009060)\** | *CTB4* | *SNOG_05808 \** |
| Monooxygenase | – | *CTB6* | – |
| Monoxygenase | – | *CTB7* | *SNOG_06495 \** |
| Hydroxylase/desaturase | – | *CTB9* | – |
| Dehydratase | – | *CTB10* | – |

\*Predicted as orthologous, but located outside the gene cluster.

genes in the perylenequinone cluster depicted in Fig 2A are significantly upregulated (Fig 2B), with expression of the core PKS gene showing near 5-fold increase in the light. The ortholog of *elcH* in *P. nodorum, g8166*, and *g9122*, predicted as orthologous to *CTB4* and encoding an MFS transporter, are both also upregulated significantly. One other gene near the cluster, *g6530*, encoding a glycosyl hydrolase (InterPro #IPR052974) was significantly upregulated, but it is not homologous with any genes in the other clusters. No other genes within 35kb of the central PKS gene were significantly upregulated.

In addition to the PKS in the perylenequinone-producing cluster, one other type-1 PKS gene, *g911*, was significantly upregulated, with a 7-fold change. Twelve consecutive genes, *g911-922*, were significantly upregulated under light. Exploration with CAGECAT revealed homologous clusters in numerous other Ascomycete fungi, some of which are known to be related to conidial or sclerotial pigmentation. One gene encoding an NRPS, *g8112*, predicted to produce isocyanide, was also significantly upregulated under light. It was the only gene upregulated within the cluster, as predicted by AntiSMASH (cluster 62.1). No secondary metabolite synthesis genes were significantly upregulated under dark conditions.

## Detection of toxin

Pigmented extracts from light- and dark-grown tissue and associated agar, along with infected leaf tissue, were identified with LC-MS/MS. Culture tissue from all isolates that were exposed to light contained both elsinochrome A and hypocrellin A. The average concentration of elsinochrome A in those samples was 718 ng/g of tissue (ranging from 39 to 1598 ng/g of tissue), while the average concentration of hypocrellin A was 28 ng/g of tissue (ranging from 8 to 76 ng/g of tissue) (Tables 3, S5 and S6). Elsinochrome A and hypocrellin A were detected at significantly higher quantities in tissue grown under light conditions compared to dark (elsinochrome A: $t_7 = 3.2$, $P < .05$; hypocrellin A: $t_7 = 2.36$, $P < .05$). Hypocrellin B was not detected in tissue grown in either the light or the dark. Significantly more elsinochrome A was detected in the agar for isolates maintained in light ($t_7 = 2.67$, $P < .05$), but concentrations were lower than in the tissue (Table 3). Only one isolate, MT.M.P17, had detectable elsinochrome A in the agar maintained in the dark (S5 Table). In the three assessed field specimens of *C. glycines*, elsinochrome A, hypocrellin A, and hypocrellin B were detected at average concentrations of

**Table 3. Mean concentration of the phytotoxic metabolites elsinochrome A and hypocrellin A in _Coniothyrium glycines_ samples under different lighting conditions.**

| Sample type | Elsinochrome A (ng/g of tissue)[1] | Hypocrellin A (ng/g of tissue)[1] |
|---|---|---|
| _C. glycines_ culture tissue in light | 616.1* | 21.5* |
| _C. glycines_ culture tissue in dark | 170.4 | 12.8 |
| _C. glycines_ culture agar in light | 11.3* | <LOD[2] |
| _C. glycines_ culture agar in dark | <LOD | <LOD |
| Diseased, field collected leaf | 9177.6 | 424.0 |
| Diseased, lab inoculated leaf | 3.8 | <LOD |

1 Treatments are compared only light vs. dark and not between samples. *Signifies $P < .05$.

2 LOD, or limit of detection is 1.5 ng/g for elsinochrome A and hypocrellins.

(9177, 424, and 103 ng/g), respectively, while lab-inoculated soybean leaves only had detectable levels of elsinochrome A at an average concentration of 3.8 ng/g. Cercosporin was not detected in the agar or tissue of _C. glycines_ in culture. One field-collected leaf specimen contained a small amount of cercosporin.

### Detached leaf assays

All isolates under both lighting treatments caused symptoms consistent with RLB, but between treatments, the average day to symptom onset and number of lesions per leaf varied significantly with some isolates (Table 4). Isolate MT.M.P17 was the only isolate where the average day to symptom onset occurred significantly sooner in the light, at 9.7 days, compared to 13.4 days, in the dark (Fig 3). For other isolates, there was no statistically significant difference. At 21 DPI, isolates MT.M.P17, MPO.P9 and NAL.3 had significantly more lesions on leaves maintained in the light (13.8–43.8 lesions per leaf) compared to the dark (4.3–18 lesions per leaf) (Table 4). For isolate EMETT.97, there was no statistically significant difference in number of lesions (Table 4).

### Discussion

Despite the regulatory importance of _C. glycines_, there are no scientific reports available describing molecular factors associated with disease. Understanding how _C. glycines_ infects plant hosts at the molecular level will be useful for developing effective management strategies. From a contemporary isolate, we identified a polyketide synthase gene with high sequence

**Table 4. Mean symptom onset and lesion occurrence in the soybean variety Williams 82 infected by four isolates of _Coniothyrium glycines_ grown under light and dark conditions.**

| Isolate | Treatment | Mean day to symptom onset (DPI) | Average number of lesions per leaf |
|---|---|---|---|
| EMETT.97 | Light | 10.8 | 10.5 |
| | Dark | 10.5 | 11.9 |
| MPO.P9 | Light | 8.5 | 27.9** |
| | Dark | 11.3 | 6.1** |
| NAL.3 | Light | 7.6 | 42.8** |
| | Dark | 8.4 | 18.0** |
| MTM.P17 | Light | 9.7* | 13.8** |
| | Dark | 13.4* | 4.3** |

Treatments are compared only light vs dark, and not between samples. *Signifies $P < .05$; ** signifies $P < .01$.

| 7 dpi | 15 dpi | 21 dpi | 28 dpi |

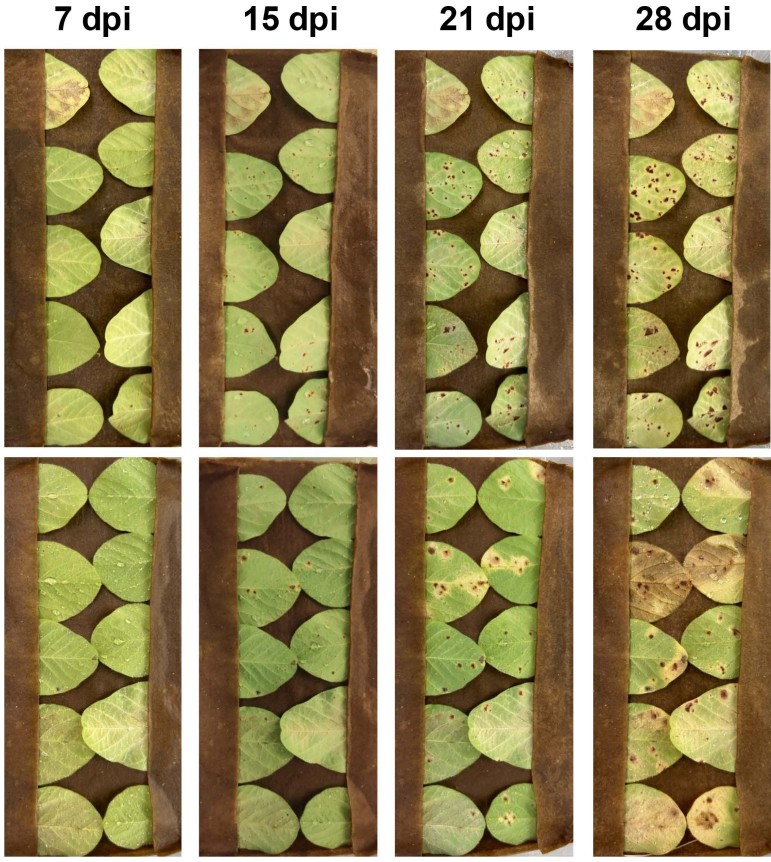

**Fig 3. Progression of disease in detached cv. Williams 82 soybean leaves inoculated with the Zambian isolate MTM.P17.** Shown at 7-, 15-, 21-, and 28-days post inoculation (DPI) under light (top) and dark (bottom) conditions. More lesions are observed in leaves incubated under light conditions, although symptoms still develop in the dark.

similarity to those that synthesize precursors to perylenequinones, a class of compounds that includes the well-studied cercosporin, elsinochrome and hypocrellin toxins, which can play major roles in plant pathogenicity [17]. We observed that a diverse selection of *C. glycines* isolates produced a pink-red pigment upon exposure of the culture to light but not when maintained in the dark, a phenotype typical of perylenequinone-producing fungi (Fig 1). Liquid chromatography with tandem mass spectrometry confirmed the production of elsinochrome A, and to a lesser extent, hypocrellin A, in cultures of *C. glycines* and field-infected leaves. Production of both compounds was significantly higher for cultures grown in light.

Phylogenetic analysis resolved the *C. glycines* PKS to a clade with those from a *Shiraia* sp. and *Parastagonospora nodorum* that are responsible for the synthesis of hypocrellin and elsinochrome C, respectively (Fig 2A). Visualization of the biosynthetic gene cluster in *C. glycines* showed the presence of eight core genes. RNA-Seq analysis showed that six of the eight genes in this cluster are co-regulated and are significantly upregulated when exposed to light (Fig 2). In *P. nodorum*, orthologs of the genes in the biosynthetic cluster synthesize prehypocrellin [66]. Prehypocrellin is the precursor to perylenequinone toxins [67]. In *C. beticola*, cercosporin is synthesized from prehypocrellin via the enzymatic activities of CTB6, CTB7, CTB9, and CTB10 [68], orthologs of which are absent in *C. glycines,* consistent with the lack of cercosporin production observed in this study.

The perylenequinone biosynthetic gene cluster in *C. glycines* is not identical in gene order or content to the other perylenequinone clusters whose products have been chemically characterized (Fig 2A and Table 2). In *P. nodorum*,

elsinochrome synthesis is catalyzed by the hydroxylase *elcH* [66], located within the cluster in *P. nodorum* (Fig 2A). While an ortholog of *elcH* is absent from the *C. glycines* biosynthetic cluster, one is present on another contig and is significantly upregulated under light, suggesting a similar metabolic pathway for elsinochrome biosynthesis in *C. glycines.* Although the LW.SEE.SI assembly is not chromosomal level, the contig with the gene cluster is well assembled, with 90kb and 120kb on either side of *g6532-g6539*, respectively. If the *elcH* ortholog is on the same chromosome, it is well outside the gene cluster. It is possible other derivatives of prehypocrellin are also produced but were not detected without a standard for comparison.

Light is usually required for lesion development with cercosporin-producing disease agents [17], and infected plants grown under low light have reduced disease ratings and delayed symptom expression [19,69]. Given this, we were surprised to observe symptom development in the *C. glycines*-inoculated leaves maintained in the dark (Fig 3). Light, however, is still a significant factor affecting lesion development. In one of the four isolates we tested, lesions on leaves maintained in the light appeared significantly earlier compared to leaves maintained in the dark, and in three of the four isolates we tested, there were significantly more lesions on leaves maintained in the light compared to the dark (Table 3, Fig 3, and S2 Table). These results are consistent with studies conducted with various *Cercospora* spp., where cercosporin-producing isolates produce a significantly greater number of lesions than non-cercosporin-producing disruption mutants [70–72]. We also detected drastically different levels of elsinochrome A and hypocrellin A between our field collected specimens and our lab-inoculated leaves (Table 3). The disparity in metabolite concentrations may be due to the higher light intensity or quality of sunlight or may indicate that toxin production increases in the later stages of infection, past the 28 days assessed in the experiment.

While most genes in the perylenequinone biosynthetic cluster were upregulated in the light, all the genes were also expressed to some degree in tissue grown in the dark (Fig 2B). Likewise, while elsinochrome A was produced in greater quantities in cultures under light conditions, some was still produced in darkness. Similarly, Chooi et al. 2017 [27] observed that elsinochrome production in *P. nodorum* also occurred in the dark. Without light, however, perylenequinones will not generate reactive oxygen species, critical to their mode of action. The occurrence of disease symptoms in the dark may be attributable to other effectors or secondary metabolites in *C. glycines* that are not induced by light. RNA-Seq analysis of *C. glycines*-infected soybean leaves will be useful to illuminate expression of genes within this biosynthetic gene cluster and those encoding proteinaceous effectors that contribute to infection of leaves maintained in light and dark.

The broad toxicity of activated oxygen species may explain the relatively wide host range of *C. glycines* in legumes in laboratory studies [73] along with the difficulty in identifying RLB-resistant soybean germplasm [2–4]. This trait complicates development of effective management strategies. One possible long-term strategy is through the identification and deployment of autoresistance genes. Secondary metabolite/toxin-producing organisms must have a mechanism in place that confers resistance to the activity of the metabolite to avoid self-harm. This capacity is termed autoresistance, which can take the form of toxin export or detoxification of toxic compounds [74]. Proteins that confer resistance to the activity of the secondary metabolite are typically located within fungal secondary metabolite gene clusters [75]. For diseases caused by species of *Cercospora*, host resistance has been achieved through transformation of the pathogen autoresistance genes to the host plant [76]. If this *C. glycines* perylenequinone toxin is confirmed to be an important virulence factor, identification of autoresistance may be a fruitful avenue of research.

In our investigation of the *C. glycines* perylenequinone gene cluster, we identified six additional fungal taxa whose genomes contain homologous gene clusters that were either poorly characterized or unknown. Besides the well-characterized clusters from *Shiraia* spp. and *P. nodorum*, the other most closely related clusters to *C. glycines* are from Phaeosphaeraceae *sp.* PMI808 and *Paraphoma chrysanthemicola*. Phaeosphaeraceae *sp.* PMI808 was isolated from the roots of a healthy *Populus* species, while *P. chrysanthemicola,* also within the Phaeosphaeriaceae in the Pleosporales, causes leaf spot diseases on species of the *Atractylodes* thistle in China [77,78]. The lesions caused by *P.*

*chrysanthemicola* appear similar to those caused by *C. glycines* [77]. As with Phaeosphaeraceae sp. PMI808, available images of *P. chrysanthemicola* in culture show formation of reddish pigments [77].

The eight genes we identified in the *C. glycines* cluster are conserved in the same order in *Melanomma pulvis-pyrius* and, remarkably, in *Corynespora cassiicola.* Both biosynthetic gene clusters also contain homologs of *elcH*, indicating an elsinochrome as a likely product. *Melanomma pulvis-pyrius* is a saprotrophic or perhaps weakly pathogenic species in the Pleosporales common on dead or decaying wood (Hashimoto et al. 2017). *Corynespora cassiicola* is a major plant pathogen of over 300 plant species, including soybean, cotton, tomato, cucumber and rubber, and causes significant economic losses [79]. It is known to produce the toxin cassiicolin, which contributes to virulence [80]. The presence of other toxins has been suspected based on the timing of symptoms and gene expression data [80]. Uncommonly, it can also cause phaeohyphomycosis, a skin disease, often of farmers [81]. Published images of some cultures of *C. cassiicola* are distinctly reddish, including those isolated from a human patient [81,82]. More distantly, we also identified perylenequinone gene clusters in two lichen-forming species within Trypetheliaceae, *Viridothelium virens* and *Bathelium mastoideum*. It was previously known that some lichens produce perylenequinones, and it is hypothesized that these metabolites offer a form of photoprotection [83].

Biosynthetic gene clusters typically have narrow taxonomic distributions [84]. In contrast, the perylenequinone biosynthetic gene cluster has been horizontally transferred to a diverse group of ascomycete plant pathogens and subsequently maintained [24]. Here we show additional, diverse taxa with complete perylenequinone synthesis clusters. Our findings underscore the likely importance of this class of toxins for the development of disease in diverse pathosystems and ecologies and suggest that they may have an underappreciated impact on agriculture.

The short list of fungi identified here as likely producers of perylenequinone toxins is far from complete, and numerous other fungi with blastp hits just below our cutoff of 60% have multiple homologous genes, based on preliminary exploration with the CAGECAT and cblaster tools. Given current trends, as more fungal genomes become available, especially within the Dothideomycetes and Pleosporales, more orthologous clusters are sure to be discovered with implications for plant disease management. The conservation of gene clusters makes identifying them within a genome relatively straightforward. For new and emerging plant pathogens, or human or animal pathogens, particularly those within the Dothideomycetes, where the molecular basis of disease remains unknown, looking for a perylenequinone biosynthetic gene cluster is a feasible first step. This may be especially recommended when cultures of Dothideomycete fungi are distinctly reddish when grown under light.

## Conclusions

This is the first known study to examine the content and expression of secondary metabolites in the Select Agent plant pathogen, *Coniothyrium glycines.* We identified a functional secondary metabolite gene cluster that is upregulated in culture upon exposure to light and leads to the production of elsinochrome A, a toxin known to contribute to virulence in other fungi. Through examination of publicly available genomes, we show that perylenequinone biosynthetic gene clusters are more widely distributed throughout the Dothideomycetes than previously known, suggesting the cluster may produce a metabolite of importance to diverse plant pathogens. In our detached leaf assays, we observed fewer lesions and later onset of disease in the leaves maintained in the dark. While this study lays the groundwork for understanding how *C. glycines* causes diseases at the molecular level, future work should include genetic knockouts to investigate the significance the gene cluster plays in pathogenicity. The development of effective management strategies to control RLB may require understanding how *C. glycines* causes disease at the molecular level.

## Supporting information

**S1 Table. Reference genomes used for gene cluster analysis.**
(XLSX)

**S2 Table. Accession numbers for RNA-Seq datasets.**
(XLSX)

**S3 Table. Annotation information for Fig 2A legend.**
(XLSX)

**S4 Table. Significantly differentially expressed genes under light or dark treatment with annotations from Interpro/Panther.**
(XLSX)

**S5 Table. Elsinochrome A concentrations (ng/g of material) in sample types.**
(XLSX)

**S6 Table. Hypocrellin A concentrations (ng/g of material) in sample types.**
(XLSX)

## Acknowledgments

The authors would like to thank Glen Hartman, Doug Luster, Sprine Misiani, Faith Sunguti and Aaron Sechler for project support, as well as express our appreciation to all who assisted with the RLB field and lab work including Clint Slocum, Godfree Chigeza, Florence Kamwana, Mathe Lukanda, Learnmore Mwadzingeni, Christabell Nachilima, Irodino Saraivia, Yechalew Sileshi, Munashe Sithole, Phinehas Tukamuhabwa, and Maserasha Yirga. This research used resources provided by the SCINet project and/or the AI Center of Excellence of the USDA Agricultural Research Service, ARS project numbers 0201-88888-003-000D and 0201-88888-002-000D. It was also supported in part by a postdoctoral fellowship funded by the USDA Agricultural Research Service's SCINet Program and AI Center of Excellence and administered by the Oak Ridge Institute for Science and Education (ORISE) through an interagency agreement between the U.S. Department of Energy (DOE) and the U.S. Department of Agriculture (USDA). ORISE is managed by ORAU under DOE contract number DE-SC0014664. All opinions expressed in this paper are the authors' and do not necessarily reflect the policies and views of USDA, DOE, or ORAU/ORISE. Mention of trade names or commercial products in this article is solely for the purpose of providing specific information and does not imply recommendation or endorsement by the U.S. Department of Agriculture (USDA). The USDA is an equal opportunity provider and employer.

## Author contributions

**Conceptualization:** Nicholas Greatens, Steven J. Clough, Michael Sulyok, Wahab Oluwanisola Okunowo, Hamed K. Abbas, W. Thomas Shier, Rachel Koch Bach.

**Data curation:** Nicholas Greatens.

**Formal analysis:** Nicholas Greatens, Michael Sulyok, Rachel Koch Bach.

**Investigation:** Nicholas Greatens, Wahab Oluwanisola Okunowo, W. Thomas Shier.

**Methodology:** Nicholas Greatens, Wahab Oluwanisola Okunowo, Rachel Koch Bach.

**Project administration:** Steven J. Clough, Hamed K. Abbas, W. Thomas Shier, Rachel Koch Bach.

**Resources:** Harun M. Murithi, Hamed K. Abbas, W. Thomas Shier.

**Supervision:** Danny Coyne, Steven J. Clough, Rachel Koch Bach.

**Visualization:** Nicholas Greatens, Rachel Koch Bach.

**Writing – original draft:** Nicholas Greatens, Rachel Koch Bach.

**Writing – review & editing:** Nicholas Greatens, Harun M. Murithi, Michael Sulyok, Wahab Oluwanisola Okunowo, Hamed K. Abbas, W. Thomas Shier, Rachel Koch Bach.

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
