## [Decision Letter · Decision Letter 0]

15 Nov 2024

PONE-D-24-43331Presence of a gene cluster for a light-activated phytotoxin in the soybean pathogen Coniothyrium glycines hints at possible virulence mechanismPLOS ONE

Dear Dr. Koch Bach,

Thank you for submitting your manuscript to PLOS ONE. After careful consideration, we feel that it has merit but does not fully meet PLOS ONE’s publication criteria as it currently stands. Therefore, we invite you to submit a revised version of the manuscript that addresses the points raised during the review process.

Please address the suggestions of the reviewers in your revised manuscripts.  I do not believe that additional experimentation is warranted.

We look forward to receiving your revised manuscript.

Kind regards,

Katherine A. Borkovich, Ph.D.

Academic Editor

PLOS ONE

“Financial support for this project was provided by USDA-ARS project 8044-22000-051-00D (RAKB and NG), USDA-ARS project 5012-22000-023-00D (SJC, and HMM) and the USDA-ARS National Plant Disease Recovery System (HKA and WTS). The authors would like to thank Glen Hartman, Doug Luster, Sprine Misiani and Aaron Sechler for project support, as well as express our appreciation to all who assisted with the RLB field and lab work including Clint Slocum, Godfree Chigeza, Florence Kamwana, Mathe Lukanda, Learnmore Mwadzingeni, Christabell Nachilima, Irodino Saraivia, Yechalew Sileshi, Munashe Sithole, Phinehas Tukamuhabwa, and Maserasha Yirga. This research used resources provided by the SCINet project and was also supported in part by a postdoctoral fellowship funded by the USDA Agricultural Research Service's SCINet Program and AI Center of Excellence, ARS project numbers 0201-88888-003-000D and 0201-88888-002-000D, and administered by the Oak Ridge Institute for Science and Education (ORISE) through an interagency agreement between the U.S. Department of Energy (DOE) and the U.S. Department of Agriculture (USDA). ORISE is managed by ORAU under DOE contract number DE-SC0014664. All opinions expressed in this paper are the authors’ and do not necessarily reflect the policies and views of USDA, DOE, or ORAU/ORISE. Mention of trade names or commercial products in this article is solely for the purpose of providing specific information and does not imply recommendation or endorsement by the U.S. Department of Agriculture (USDA). The USDA is an equal opportunity provider and employer.”

“Financial support for this project was provided by USDA-ARS project 8044-22000-051-00D (RAKB and NG) and the USDA-ARS National Plant Disease Recovery System (HKA and WTS).”

Reviewers' comments:

Reviewer's Responses to Questions

**Comments to the Author**

1. Is the manuscript technically sound, and do the data support the conclusions?

Reviewer #1: Yes

Reviewer #2: Partly

2. Has the statistical analysis been performed appropriately and rigorously? 

Reviewer #1: Yes

Reviewer #2: Yes

3. Have the authors made all data underlying the findings in their manuscript fully available?

Reviewer #1: Yes

Reviewer #2: Yes

4. Is the manuscript presented in an intelligible fashion and written in standard English?

Reviewer #1: Yes

Reviewer #2: Yes

5. Review Comments to the Author

Reviewer #1: The authors carried out a study to investigate the presence of a secondary metabolite gene cluster in the genome of C. glycines, and how exposure to light/darkness affected pigment production in axenic cultures and symptom development in detached leaves. The study is timely and the manuscript is well written.

Minor correction:

Line 289: Indicate that the counts of genes g6532-g6539 are from Coniothyrium glycines.

Reviewer #2: This manuscript describes potential perylenequinone production from Coniothyrium glycines, an important soybean pathogen. The authors showed via whole genome sequencing that this species has a secondary metabolite cluster very closely related to those that produce cercosporin; a well-studied secondary metabolite effector produced by Cercospora species and other Dothidiomycetes. Isolates of C. glycines were grown in light/dark conditions, which showed a reddish phenotype in the mycelium potentially characteristic of cercosporin in the light-grown cultures, which was absent in the dark-grown cultures. The authors showed that several of the genes in this cluster were up-regulated in the light, also typical of cercosporin. The authors used a detached leaf assay to show that there was less disease when inoculated leaves were in the dark versus the light, suggesting that a light-activated effector is at play with this fungus.

I found the manuscript to be well-written and a pretty compelling argument for perylenequinone production by this fungus, and likely cercosoporin. This manuscript, however, relies heavily on association (especially to phenotypes) without development of mutants, etc to really show that this fungus is producing cercosporin. The detached leaf assay is somewhat compelling, but there could be a host of other reasons why there was less disease in the dark. However, with a little extra lab work, cercosporin production can be shown very easily (see below) which will help add impact to this work.

Points to consider:

Figure 1 cultures show light/day-related color and morphology, but is not diagnostic for cercosporin. In fact, cercosporin should be secreted into the medium in advance of the mycelium… typically.

The “eight gene cluster” of cercosporin is mentioned in several cases; it is not produced via only eight genes (see de Jonge et al PNAS 2018). The bottom C. beticola genome should really be removed in figure 2 as it doesn’t add anything to the figure.

Figure captions (and Tables) within the main text itself makes reading the manuscript difficult.

Table 2: C. beticola has at least 13 CTB genes, including MFS transporter… Why aren’t the others included here?

Finally, consider the standard “KOH Assay” for cercosporin detection:

1. Grow isolates on 9 cm Petri plates containing 15 ml of PDA

a. 7d, 25C, Natural light

2. Using a #2 cork borer, remove three plugs from each isolate from the edge, middle and center of each colony and placed in small screw cap glass vials. (scrape off excess PDA)

3. Add 5mls of 5N KOH to each vial to cover the surface of the plugs and shake at room temperature for 4 hours (this can be shortened more like 15-30min)

4. Supernatants were examined for cercosporin spectrophotometrically.

5. Spectrophotometer set to broad spectrum using 472 nm as the target wavelength.

Cercosporin (Sigma) can be used as a positive control in this assay (dissolve in acetone to 100 mM)

6. PLOS authors have the option to publish the peer review history of their article (what does this mean? ). If published, this will include your full peer review and any attached files.

**Do you want your identity to be public for this peer review?** For information about this choice, including consent withdrawal, please see our Privacy Policy .

Reviewer #1: **Yes: ** Emma W. Gachomo

Reviewer #2: No

---

## [Author Response · Author response to Decision Letter 1]

11 Mar 2025

Editor comments:

“Financial support for this project was provided by USDA-ARS project 8044-22000-051-00D (RAKB and NG), USDA-ARS project 5012-22000-023-00D (SJC, and HMM) and the USDA-ARS National Plant Disease Recovery System (HKA and WTS). The authors would like to thank Glen Hartman, Doug Luster, Sprine Misiani and Aaron Sechler for project support, as well as express our appreciation to all who assisted with the RLB field and lab work including Clint Slocum, Godfree Chigeza, Florence Kamwana, Mathe Lukanda, Learnmore Mwadzingeni, Christabell Nachilima, Irodino Saraivia, Yechalew Sileshi, Munashe Sithole, Phinehas Tukamuhabwa, and Maserasha Yirga. This research used resources provided by the SCINet project and was also supported in part by a postdoctoral fellowship funded by the USDA Agricultural Research Service's SCINet Program and AI Center of Excellence, ARS project numbers 0201-88888-003-000D and 0201-88888-002-000D, and administered by the Oak Ridge Institute for Science and Education (ORISE) through an interagency agreement between the U.S. Department of Energy (DOE) and the U.S. Department of Agriculture (USDA). ORISE is managed by ORAU under DOE contract number DE-SC0014664. All opinions expressed in this paper are the authors’ and do not necessarily reflect the policies and views of USDA, DOE, or ORAU/ORISE. Mention of trade names or commercial products in this article is solely for the purpose of providing specific information and does not imply recommendation or endorsement by the U.S. Department of Agriculture (USDA). The USDA is an equal opportunity provider and employer.”

“Financial support for this project was provided by USDA-ARS project 8044-22000-051-00D (RAKB and NG) and the USDA-ARS National Plant Disease Recovery System (HKA and WTS).”

We have included the amended statement in the cover letter. It reads as the following: Our funding sources are as follows: USDA-ARS projects 8044-22000-051-00D, 5012-22000-023-00D, 6066-42000-007-000D, 0201-88888-003-000D, and 0201-88888-002-000D, as well as the USDA-ARS National Plant Disease Recovery System.

Reviewer #1:

The authors carried out a study to investigate the presence of a secondary metabolite gene cluster in the genome of C. glycines, and how exposure to light/darkness affected pigment production in axenic cultures and symptom development in detached leaves. The study is timely and the manuscript is well written.

Minor correction:

Line 289: Indicate that the counts of genes g6532-g6539 are from Coniothyrium glycines.

Accepted

Reviewer #2: This manuscript describes potential perylenequinone production from Coniothyrium glycines, an important soybean pathogen. The authors showed via whole genome sequencing that this species has a secondary metabolite cluster very closely related to those that produce cercosporin; a well-studied secondary metabolite effector produced by Cercospora species and other Dothidiomycetes. Isolates of C. glycines were grown in light/dark conditions, which showed a reddish phenotype in the mycelium potentially characteristic of cercosporin in the light-grown cultures, which was absent in the dark-grown cultures. The authors showed that several of the genes in this cluster were up-regulated in the light, also typical of cercosporin. The authors used a detached leaf assay to show that there was less disease when inoculated leaves were in the dark versus the light, suggesting that a light-activated effector is at play with this fungus.

I found the manuscript to be well-written and a pretty compelling argument for perylenequinone production by this fungus, and likely cercosoporin. This manuscript, however, relies heavily on association (especially to phenotypes) without development of mutants, etc to really show that this fungus is producing cercosporin. The detached leaf assay is somewhat compelling, but there could be a host of other reasons why there was less disease in the dark. However, with a little extra lab work, cercosporin production can be shown very easily (see below) which will help add impact to this work.

Points to consider:

Figure 1 cultures show light/day-related color and morphology, but is not diagnostic for cercosporin. In fact, cercosporin should be secreted into the medium in advance of the mycelium… typically.

With our additional LC-MS/MS data, we were able to test this out and we found that not much of the perylenequinone elsinochrome is secreted into the agar, but is instead concentrated in the tissue. Perhaps the different chemical properties of cercosporin and elsinochrome may explain the difference.

The “eight gene cluster” of cercosporin is mentioned in several cases; it is not produced via only eight genes (see de Jonge et al PNAS 2018).

The eight gene cluster refers to the eight genes present in C. glycines that form prehypocrellin, a precursor to perylenequinones, such as elsinochromes. Hu et al. 2019 show these genes, when expressed in Aspergilllis nidulans in absence of other enzymes that modify prehypocrellin, form hypocrellins. We acknowledge the other genes involved in cercosporin biosynthesis in Fig 2, and, following recommended edits below, in Table 2.

The bottom C. beticola genome should really be removed in figure 2 as it doesn’t add anything to the figure.

Accepted. The gene visualization has been removed, but the PKS gene has been retained for use as an outgroup in the tree.

Figure captions (and Tables) within the main text itself makes reading the manuscript difficult.

Captions are now reformatted to comply with PLOSONE requirements. Hopefully, this makes reading the manuscript easier.

Table 2: C. beticola has at least 13 CTB genes, including MFS transporter… Why aren’t the others included here?

Accepted. Other CTB genes have been added for comparison. The MFS transporter in the C glycines cluster is not predicted as orthologous to the CTB MFS transporter. Both are now included in the table.

Finally, consider the standard “KOH Assay” for cercosporin detection:

1. Grow isolates on 9 cm Petri plates containing 15 ml of PDA

a. 7d, 25C, Natural light

2. Using a #2 cork borer, remove three plugs from each isolate from the edge, middle and center of each colony and placed in small screw cap glass vials. (scrape off excess PDA)

3. Add 5mls of 5N KOH to each vial to cover the surface of the plugs and shake at room temperature for 4 hours (this can be shortened more like 15-30min)

4. Supernatants were examined for cercosporin spectrophotometrically.

5. Spectrophotometer set to broad spectrum using 472 nm as the target wavelength.

Cercosporin (Sigma) can be used as a positive control in this assay (dissolve in acetone to 100 mM)

In response to reviews, we used a comparable technique with lc-ms to analyze the toxin produced by the cluster and found the primary toxin to be elsinochrome, rather than cercosporin or hypocrellin, although the latter is produced to a lesser extent. Cercosporin is not produced.

6. PLOS authors have the option to publish the peer review history of their article (what does this mean?). If published, this will include your full peer review and any attached files.

Do you want your identity to be public for this peer review? For information about this choice, including consent withdrawal, please see our Privacy Policy.

Reviewer #1: Yes: Emma W. Gachomo

Reviewer #2: No

---

## [Editor Report · Decision Letter 1]

14 Mar 2025

Production of the light-activated elsinochrome phytotoxin in the soybean pathogen Coniothyrium glycines hints at virulence factor

PONE-D-24-43331R1

Dear Dr. Koch Bach,

We’re pleased to inform you that your manuscript has been judged scientifically suitable for publication and will be formally accepted for publication once it meets all outstanding technical requirements.

Kind regards,

Katherine A. Borkovich, Ph.D.

Academic Editor

PLOS ONE
---

## [Editor Report · Acceptance letter]

PONE-D-24-43331R1

PLOS ONE

Dear Dr. Koch Bach,

I'm pleased to inform you that your manuscript has been deemed suitable for publication in PLOS ONE. Congratulations! Your manuscript is now being handed over to our production team.

Kind regards,

on behalf of

Dr. Katherine A. Borkovich

Academic Editor

PLOS ONE